# Surveillance and Coinfection Dynamics of Infectious Bronchitis Virus and Avian Influenza H9N2 in Moroccan Broiler Farms (2021–2023): Phylogenetic Insights and Impact on Poultry Health

**DOI:** 10.3390/v17060786

**Published:** 2025-05-30

**Authors:** Rim Regragui, Oumayma Arbani, Nadia Touil, Khalid Bouzoubaa, Mohamed Oukessou, Mohammed El Houadfi, Siham Fellahi

**Affiliations:** 1Department of Veterinary Pathology and Public Health, Institut Agronomique et Vétérinaire Hassan II, Rabat 10000, Morocco; arbanioumayma@iav.ac.ma (O.A.); elhouadfimohammed@yahoo.fr (M.E.H.); s.fellahi@iav.ac.ma (S.F.); 2Cell Culture Unit, Center of Virology, Infectious and Tropical Diseases, Mohammed V Military Training Hospital, Rabat 10000, Morocco; ntouil2003@gmail.com; 3Aviculture Diagnostic Center, Temara, Rabat 10000, Morocco; kbouz81@gmail.com; 4Department of Veterinary, Biological and Pharmaceutical Sciences, Institut Agronomique et Vétérinaire Hassan II, Rabat 10000, Morocco; m.oukessou@iav.ac.ma

**Keywords:** IBV, H9N2, MG, MS, ILT, coinfections, respiratory

## Abstract

Infectious bronchitis virus (IBV) and low-pathogenic avian influenza virus (LPAIV) H9N2 are commonly identified in poultry, individually or in association with other pathogens. This study monitored 183 broiler farms affected by respiratory diseases across seven regions of Morocco from January 2021 to December 2023. Among these farms, 87.98% were vaccinated against IBV, while 57.92% were against AI H9N2. Abnormally high mortality rates were observed in 44.26% of the farms, with 24.69% of cases attributed to IBV, 50.62% to LPAI H9N2, and 13.58% due to coinfection with both IBV and H9N2. RT-PCR analysis of tissue samples and cloacal and tracheal swabs collected from 183 broiler farms revealed that 33.33% were positive for IBV and 34.97% for H9N2. Coinfection by IBV and H9N2 was detected in 12.57% of cases, peaking at 17% in 2022. Co-infected flocks exhibited severe clinical signs and lesions, such as reduced food consumption, diarrhea, and renal issues. The predominant lesions were in the respiratory tract, affecting 91.26% of infected broilers. Additionally, among the 183 flocks, 50 farms that tested positive for IBV infection were randomly selected from the seven regions of Morocco for further investigation of other respiratory pathogens, including *Mycoplasma gallisepticum* (MG), *Mycoplasma synoviae* (MS), and infectious laryngotracheitis (ILT), using real-time RT-PCR. Detection rates for these pathogens were 26% for MG, 30% for MS, 4% for ILTv (vaccine strain), and 18% for ILTw (wild strain). Detection rates for single, dual, triple, and quadruple infections were 34%, 42%, 18%, and 4%, respectively. The most common dual and triple coinfections were IBV + H9N2 (14%) and IBV + MG + MS (10%). Phylogenetic analysis of the S gene identified two main IBV genotypes, namely, 793B and D181, with the latter being a strain circulating for the first time in Moroccan poultry. This underscores the urgent need to establish surveillance systems to track pathogen circulation and implement strategies to control virus spread, ensuring the protection of animals and public health.

## 1. Introduction

Avian infectious bronchitis (IB) is a viral, endemic, and highly contagious disease primarily affecting chickens. It causes respiratory, renal, and reproductive diseases and engenders significant economic losses to the poultry industry worldwide [1]. In broilers, financial losses are mainly attributed to a weight gain decrease and a high mortality rate, especially in infection with nephropathogenic strains. In layers, the main signs include an egg production decrease and a deterioration in internal and external egg quality [2]. The infectious bronchitis virus (IBV) is a single-stranded RNA virus of the *Coronaviridae* family that encodes four structural proteins: spike (S), membrane (M), envelope (E), and nucleoprotein (N) [3,4,5,6,7]. IBV is prone to changes through spontaneous mutations or genetic recombination, leading to the emergence of new variants [8]. In Morocco, the first isolation and characterization of IBV were reported in 1983 by El Houadfi and Jones. The “Moroccan G strain” was identified in 1986, and phylogenetic studies suggested a shared origin with the 4/91 serotype [9]. Between 1996 and 2005, three IBV genotypes were identified: Massachusetts (Mass.), a strain related to 4/91, and a nephropathogenic variant [10]. In 2005, five nephropathogenic IBV strains were identified in Morocco [11]. From 2009 to 2014, novel genotypes emerged, including Italy 02, which made up 32% of isolates in this period [12]. Since 2015, a new IBV lineage has been detected in different regions of Morocco, with related viruses observed in Algeria in 2012 and Tunisia in 2016 [13]. The poultry disease status in Morocco has worsened since 2016 following the introduction of AI H9N2 and its ability to cause disease alone or in combination with other pathogens [14].

LPAIV H9N2 is a highly infectious pathogen that affects birds and humans [15]. Due to its potential to reassort with other influenza viruses and to evolve into a more threatening strain, monitoring farms for H9N2 has become crucial to assess the virus’s spread and its evolution. Respiratory complex diseases often have a multifactorial nature, as more than one pathogen can be detected in the same flock, such as infectious bronchitis virus (IBV), low pathogenic avian influenza virus (LPAIV) H9N2, infectious laryngotracheitis (ILT), Newcastle disease virus (NDV), *Mycoplasma gallisepticum* (MG), *Mycoplasma synoviae* (MS), and *Escherichia coli* (E. coli). Coinfections between viral agents or viral and bacterial pathogens are common in poultry farms, often resulting in more severe clinical symptoms than single infections [16].

This study aims to (i) determine the prevalence of IBV and LPAI H9N2 in Moroccan poultry farms, (ii) identify the current IBV genotypes and lineages circulating in the country, and (iii) investigate the correlations between different avian pathogens in respiratory coinfection diseases.

## 2. Materials and Methods

### 2.1. Field Sampling and Data Collection

From January 2021 to December 2023, 183 poultry farms that experienced respiratory disease outbreaks were investigated to determine the primary causes of the respiratory outbreaks in broiler chickens. The investigations included data collection regarding bird age, clinical signs, vaccination status, mortality rates, post-mortem examinations, tissue samples, and tracheal and cloacal swabs. Sampling targeted chickens over three weeks old that showed reduced food consumption and respiratory signs such as coughing, sneezing, rales, excessive lacrimation, and rattling. Five to ten samples were collected from each flock, including tracheal swabs from live birds and tissue samples from various organs (trachea, lungs, kidneys, liver, spleen, and proventriculus) of freshly dead birds exhibiting macroscopic lesions. A total of 856 organ samples and tracheal swabs were collected from poultry farms across seven regions of Morocco (Figure 1). The samples were placed on ice and transported in a sterile medium containing 5% antibiotics (penicillin 20,000 U/mL, streptomycin 10,000 µg/mL, and kanamycin 5000 µg/mL). They were then subjected to real-time RT-PCR for the detection of IBV and H9N2. Among the 183 farms, 50 that tested positive for IBV infection were selected for further investigation of coinfection with *Mycoplasma gallisepticum* (MG), *Mycoplasma synoviae* (MS), and infectious laryngotracheitis (ILT) strains, including both vaccine (ILTv) and wild (ILTw) strains.

### 2.2. RNA/DNA Extraction and Real-Time RT-PCR

Viral RNA was extracted from the samples using the Kylt^®^ RNA/DNA Purification Kit following the manufacturer’s instructions (AniCon Labor GmbH Laboratory, Emstek, Germany). Specific primers and probes to each pathogen were used, including IBV, H9N2, MG, MS, and ILT (Table 1). The amplification was carried out using the Invitrogen™ SuperScript™ III One-Step RT-PCR Kit (Thermo Fisher Scientific, Göteborg, Sweden) on a 7500 Real-Time PCR System (Applied Biosystems, Foster City, CA, USA). The reaction volume for each pathogen was as follows: for both IBV and H9N2, the reaction mix contained 12.5 µL of 2× RT-PCR buffer mix, 1 µL of each primer (10 µM), 0.5 µL (10 µM) of probe, 0.5 µL of RT, 5 µL of RNA template, and 4.5 µL of distilled water.

Regarding ILT amplification, the reaction volume contained 12.5 µL of 2x reaction mix, 1 µL of each primer (10 µM), 0.5 µL (10 µM) of probe, 5 µL of RNA template, and 5 µL of distilled water. The reaction had specific cycling conditions for each pathogen:

IBV: 50 °C for 5 min, followed by 95 °C for 2 min, and 40 cycles of 95 °C for 3 s and 58 °C for 30 s;

For H9N2: 50 °C for 15 min, followed by 95 °C for 2 min, and 40 cycles of 95 °C for 15 s and 50 °C for 30 s;

ILT: 95 °C for 2 min, followed by 40 cycles of 95 °C for 15 S and 60 °C for 30 s.

### 2.3. Detection of MG and MS

A commercial PCR kit was used for MG and MS detection (ADL01Y1—ADIALYO™ MS/MG), with the following cycling conditions: 45 °C for 10 min, followed by 95 °C for 2 min, and 40 cycles of 95 °C for 5 s and 60 °C for 30 s.

### 2.4. IBV Isolation

IBV isolation was attempted from all IBV-positive samples using real-time RT-PCR. Briefly, 200 µL of tissue homogenate or tracheal swab was inoculated into 9–11-day-old embryonated specific pathogen-free chicken eggs and incubated at 37 °C, with daily candling. Embryos that died within 24 h post-inoculation were discarded. The remaining eggs were incubated for seven days. The allantoïc fluids collected were then passaged three times to detect IBV-induced lesions in the embryos, such as dwarfing, curling, and stunting. The allantoïc fluids from positive embryos were stored at −80 °C until further use.

### 2.5. Conventional RT-PCR and cDNA Purification

Complementary DNA (cDNA) was synthesized from extracted RNA through reverse transcription using the Applied Biosystems Kit (Life Technologies). PCR amplification was performed with the Dream Taq Green PCR Master Mix (×2) Kit (Thermo Fisher Scientific, Waltham, MA, USA) using specific primers for S1 gene amplification following the protocols described by Jones et al. (2005) [9]. Partial RT-PCR products of the S1 gene were purified using the Gene Clean Kit (ExoSAP-IT, Affymetrix, Santa Clara, CA, USA) following the manufacturer’s instructions.

### 2.6. DNA Sequencing and Phylogenetic Analysis of IBV Isolates

The 15 purified PCR products (392 bp) served as templates for sequencing using the BigDye^®^ Terminator v1.1 Cycle Sequencing Kit (Life Technologies, Carlsbad, CA, USA). A second purification step was performed using the Big Dye^®^ XTerminator Purification Kit (Life Technologies, Carlsbad, CA, USA). The purified PCR products were then sequenced from both directions using the same primers (SX3+ and SX4-).

Sequence data were assembled and analyzed with BioEdit software (version 7.2) [22]. Sequence comparisons with reference strains from the GenBank database were performed using the open-source BLAST program (National Center for Biotechnology Information, Bethesda, MD, USA, http://blast.ncbi.nlm.nih.gov/Blast.cgi, accessed on 22 December 2023). For the alignment of nucleotide sequences and inferred amino acid sequences, ClustalW and MEGA software (version 6.0) were employed [23]. Phylogenetic analysis and tree construction for the partial S1 glycoprotein gene sequences were conducted using the neighbor-joining method with 1000 bootstrap replicates in MEGA.

## 3. Results

### 3.1. Field Investigations

The 183 monitored farms had chickens aged between 17 and 60 days, with a median age of 36 days. Among these farms, 161 (87.98%) had implemented vaccination against IBV, while 106 (57.92%) were vaccinated against H9N2. The most prevalent clinical signs observed included abnormal mortality in 44.26% of farms and respiratory symptoms such as sneezing, coughing, and rales in 83.61% of farms. Additionally, reduced food consumption, digestive disturbances, diarrhea, and renal issues were reported. Lesions were predominantly found in the respiratory tract, affecting 91.26% of infected broilers. Kidney congestion and enteritis were also observed in 16.39% and 39.34% of cases, respectively.

### 3.2. IBV Detection

Among the 183 surveyed farms, 61 (33.33%) tested positive for IBV. Among these IBV-positive farms, 47 (77.05%) had previously been vaccinated against IBV. The yearly distribution of IBV detection was highest in 2021 (50.94%), declined in 2022 (33%), and dropped abruptly in 2023 (3.33%). Non-vaccinated farms recorded the highest positivity rate (78.57%), while those that were IBV-vaccinated recorded a positivity rate of 53.45%. In contrast, farms that received the Mass-type vaccine and H9N2 inactivated vaccines had a significantly lower rate of 15.53%. Regarding age, broilers aged between 41 and 60 days had the highest IBV detection rate (37.04%) compared to those aged between 21 and 40 days (14.55%) and those ≤20 days (16.67%).

### 3.3. H9N2 Detection

Among the 183 surveyed farms, 64 (34.97%) tested positive for H9N2. Among these, 37 (57.81%) had been vaccinated against LPAI H9N2. The yearly distribution for H9N2 detection peaked in 2022 (44%), while 2023 and 2021 recorded 36.67% and 18.87% of cases, respectively. Non-vaccinated farms had the highest positivity rate (71.43%), followed by those vaccinated with the H9N2 vaccine (66.67%). Farms that received the IBV Mass-type vaccine and H9N2 vaccines had a significantly lower positivity rate of 33.98%. Regarding age, H9N2 detection was highest in broilers ≤20 days (66.67%), followed by those ages 21–40 days (36.36%) and 41−60 days (25.93%).

### 3.4. Coinfection of IBV and H9N2

#### 3.4.1. Detection Rates and Age Distribution

Among the 183 surveyed farms, 23 (12.57%) tested positive for IBV and LPAI H9N2. A total of 61 farms (33.33%) were positive for IBV, while 64 (34.97%) tested positive for H9N2. Out of the 183 farms surveyed, only 88 provided age data for their broiler chickens. The median age of the broilers in this study was 36 days, with a minimum age of 17 days. Weekly mortality rates ranged from 0.1% to 16.4%. The detection rates for IBV and H9N2 were higher in broiler chickens aged 21–40 days than those aged ≤20 days or >40 days (Table 2 and Figure 2).

#### 3.4.2. Yearly Distribution

This study was carried out from 2021 to 2023. Coinfections with both IBV and H9N2 were most frequent in 2022, with a rate of 17%. In 2021 and 2023, the rates were lower, accounting for 9.43% and 3.33%, respectively (Table 3 and Figure 3).

#### 3.4.3. Vaccination Status

Of 183 farms, 161 were vaccinated against IBV and 106 against H9N2, representing 87.98% and 57.92%, respectively. Among the 61 IBV-positive farms, 47 had been vaccinated (77.05%), while among the 64 H9N2-positive farms, 37 had been vaccinated (57.81%) (Table 4 and Figure 4).

#### 3.4.4. Phylogenetic Analysis of IBV (D181 and 793B) Strains

The phylogenetic tree was constructed based on the partial S1 gene sequence (392 bp) using the neighbor-joining method with 1000 bootstrap replicates (Figure 5). The analysis showed that our Moroccan isolates were grouped into two major clusters corresponding to the 793B and D181 lineages.

The first cluster (793B genotypes) included IBV/Morocco/2021/P173-1, P173-2, P173-3, IBV/Morocco/2022/P177-1, P177-2, P178-1, P179-2, P172-1, and P172-2, showing nucleotide identities ranging from 98.5% to 99.6% with the 4/91 vaccine strain (MF156213.1 Morocco 4/91 IBV02 2014). These isolates clustered closely with UK 4/91 (JN192154.1) and Spain 92/51 (DQ064801.1) by high bootstrap values (93–100).

The second cluster (D274 genotypes) included IBV/Morocco/2022/P161(1), P160, P164(Ac), P164(T), and P165(Ac), with nuclear identities ranging from 97.9% to 99.2%. These strains clustered closely with the Dutch strains D181 (MK840961.1 CK/NL/D181/2018) and D1466 (MN548286.1), with a high bootstrap support (100). These isolates’ classification is within the D74 genotype (D181-like strains).

No Moroccan strains grouped with the QX, D274, Gray, Ark99, M41, Beaudette, Ma5, H120, or H52 strains, indicating no introduction or dominance of these genotypes in the sampled Moroccan population.

### 3.5. Other Coinfections

Out of the 183 surveyed farms, 50 farms positive for IBV infection were selected for other pathogen coinfections including MG, MS, and ILT (ILTv and ILTw).

#### 3.5.1. ILT Results

From 2021 to 2023, ILT was detected in three regions: Rabat-Sale-Kenitra, Casablanca-Settat, and Marrakech-Safi. A total of seven cases were recorded in vaccinated farms, including three coinfections with IBV, two coinfections with H9N2, one case of coinfection with IBV and H9N2, and one case with simultaneous infection of both ILTv and ILTw.

#### 3.5.2. MG and MS Results

From 2021 to 2023, 13 of 50 farms were positive for MG, and 15 were positive for MS in Rabat-Sale-Kenitra, Casablanca-Settat, and the East. In vaccinated farms, four were positive for IBV and MS, two for MG and MS, and one case of IBV, H9N2, and MS was recorded. In unvaccinated farms, one case of IBV and MS and one case of IBV, H9N2, and MG were recorded.

#### 3.5.3. Detection Rates of the Respiratory Pathogens

In the 183 farms surveyed, the detection rate for single infections was 37.16%, while coinfections were detected in 22.95% of cases. Among these farms, 23 were co-infected with IBV and H9N2, accounting for 12.57%. Additionally, 50 farms were randomly tested for MG, MS, and ILT (ILTv and ILTw). The detection rates for single, dual, triple, and quadruple infections were 34%, 42%, 18%, and 4%, respectively. The most common dual and triple infections were IBV and H9N2 (14%) and IBV, MG, and MS (10%) (Figure 6).

#### 3.5.4. Correlation Between Number of Coinfections and Vaccination Protocols

Among the vaccinated farms, 55.56% (25/45) tested positive for more than one virus. The most frequently detected pathogens were IBV, H9N2, and MS. Seven cases of triple infections were observed, all involving IBV, MG, and MS, representing 28% (7/25). A quadruple infection (IBV, H9N2, MG, and MS) was also recorded (Table 5). Similarly, all five non-vaccinated farms were infected, with two cases each of dual and triple infections and one quadruple infection. These infections consistently involved the same recurrent pathogens: IBV, H9N2, MG, and MS (Table 6). All coinfection combinations are presented in Figure 7.

#### 3.5.5. Correlation Between Age, Infection, and Vaccination Protocol

The median age of the 50 farms tested was 38 days, with a minimum age of 22 days old. Seventeen infections were recorded in vaccinated chickens aged 20–40 days, with infection peaks at 24 and 38 days, indicating a higher susceptibility in this age range. An additional 18 cases were recorded in vaccinated chickens aged 40–60 days, with a peak at 54 days. Both younger and older broilers showed infection trends, with similar case numbers in each age group.

#### 3.5.6. Regional Distribution of Pathogens

Among the seven regions of Morocco, four accounted for 60% (30/50) of coinfection cases, with Rabat-Sale and Casablanca-Settat showing the highest detection rates, at 50% (15/30) and 40% (12/30), respectively. The highest quadruple coinfections cases were recorded in the Casablanca-Settat and Rabat-Sale-Kenitra regions. The three-pathogen combinations were prevalent in the Casablanca-Settat, Rabat-Sale-Kenitra, and Marrakech-Safi regions. The dual coinfections, however, were registered across four regions of Morocco (Table 7).

## 4. Discussion

Infectious bronchitis virus (IBV) and low-pathogenic avian influenza virus (LPAIV) H9N2 subtype are high-risk pathogens posing serious threats to animal health, public health, and the economy. The World Organization for Animal Health (WOAH) identifies both viruses as significant dangers to the poultry industry and public health due to the outbreak potential of IBV and the zoonotic risk of H9N2 [24,25]. This study investigated the prevalence of IBV and H9N2 in 183 broiler farms across seven regions of Morocco from 2021 to 2023. The detection rates were 33.33% for IBV and 34.97% for H9N2, which were nearly equivalent, suggesting that common risk factors facilitate their spread. Previous studies reported higher prevalence rates, with 51.7% of the flocks testing positive for IBV and 58% for H9N2 [12,26]. The coinfection rate was 12.57%, with a peak of 17% observed in 2022. The same year, H9N2 cases peaked at 44%, indicating increased viral circulation. Yearly variations in infection rates have been observed in other studies. They could be attributed to various reasons, including differences in biosecurity and control measures, poultry density, and the hygienic conditions of the studied farms [27,28]. Despite extensive vaccination coverage, 77.05% of vaccinated farms tested positive for IBV and 57.81% for H9N2. Positive cases showed high mortality rates and respiratory symptoms, with predominant respiratory lesions including congestive tracheitis, airsacculitis, and sinusitis. Tracheitis is a common clinical sign of IBV infections due to the viral damage of ciliated epithelial cells [29]. H9N2 also affects the respiratory tract and is characterized by airsacculitis, tracheitis, and sinus enlargement [30]. Broilers aged 21–40 days were particularly susceptible to these coinfections due to their immature immune systems, which may be further compromised by stress and suboptimal environmental conditions [31].

Field data showed an exacerbation of clinical signs and gross lesions in cases co-infected with IBV and H9N2, demonstrating that mixed infections enhance pathogenicity and increase mortality rates, reaching 44.26%, causing substantial economic losses [32,33,34]. Furthermore, malpractices, inadequate security measures, and/or stress factors could affect the weekly mortality rate, ranging from 0.1% to 16.4% [35].

However, the infection sequence may contribute to pathogenic differences [36,37]. Several studies have shown that coinfection with IBV and H9N2 enhances clinical signs, irrespective of the IBV strain involved, and extends the shedding period of H9N2 [37,38,39,40]. This synergistic effect is thought to result from the IBV-induced severe inflammatory response, which increases the pathogenicity of H9N2 [36]. It could also result from the presence of a trypsin-like serine protease encoded by IBV, which facilitates the cleavage activation of H9N2 hemagglutinin, thereby improving its replication [41,42]. In contrast, in vitro and ovo studies reported different results, indicating viral interference that was not observed in vivo. The quantitative results of simultaneous coinfections with H9N2 and IBV revealed that interference between the two viruses led to a reduction in viral growth. However, in cases of superinfection, the second virus—whether H9N2 or IBV—reduced the growth of the initially inoculated virus [38]. These findings may inform future strategies for virus control and the development of appropriate vaccination protocols.

To further explore potential coinfections with other pathogens, 50 farms positive for IBV were randomly selected from the 183 previously studied. These farms were tested for other respiratory pathogens. Single infections were more common, detected in 37.16% of cases, whereas coinfections had a detection rate of 22.95%. The most frequent combinations were IBV + H9N2 and IBV + MS for dual infections, IBV + H9N2 + MG and IBV + MG + MS for triple infections, and IBV + H9N2 + MG + MS for a notable quadruple infection. Therefore, IBV emerged as the predominant virus that facilitated the introduction of other pathogens. A similar study conducted by Jbenyeni et al. from 2018 to 2020 in Tunisia showed a high prevalence of IBV and LPAIV H9N2 in poultry farms, with the same major viral combination of IBV + H9N2, stressing H9N2’s contribution to multifactorial respiratory diseases [43]. In Ghana, the detection of H9N2 was frequently associated with IBV detection, causing an exacerbation of clinical signs [44].

Regarding the geographic distribution of pathogens in Morocco, Rabat-Sale-Kenitra registered the highest diversity of infections, with IBV + H9N2 being the most common dual infection. Casablanca-Settat also recorded several cases involving IBV, MG, and MS. Additionally, quadruple infections were detected in these regions, indicating their status as hotspots for viral activity and numerous pathogens. The high prevalence observed in these two regions is likely attributed to the high poultry activity due to the growing industry in these areas. Arbani et al. have reported a high prevalence of LPAI H9N2 in the same regions [45]. In contrast, fewer cases were reported in Marrakech-Safi, while only one was detected in the Eastern region, suggesting regional variation in infection dynamics.

Another notable pathogen detected in our study was ILT. While infections involving this virus were less frequent, its presence in dual and triple infections across three regions suggested ILT may synergistically exacerbate respiratory symptoms. Several studies have demonstrated that coinfections contribute to enhanced respiratory signs and increased mortality rates [46,47]. For example, chickens co-infected with IBV and H9N2 showed more severe clinical signs. Similarly, a study of the effect of LPAIV H3N8 and MG coinfection found an increased MG colonization rate in lungs and internal organs, which resulted in higher mortality rates [46]. These findings were supported by a challenge study using H9N2 and MG, where co-inoculation caused increased tracheal cast formation and higher mortality [48]. Another study highlighted the synergistic role of MS and IBV in enhancing inflammation in the respiratory tract and exacerbating lesions, particularly in airsacculitis syndrome [49].

The phylogenetic analysis showed that all IBV viruses isolated belonged to two distinct genotypes: 793B (GI-13) and D181 (GII-1). The homology analysis showed that the 793B strain was similar to the Spain/92/51, Morocco/4-91/IBV02/2014, and Morocco/4-91/IBV16/2014 strains, and the sequence identity ranged from 99% to 100%.

First isolated in 1985 in France [50], 793B entered the United Kingdom in 1991 to finally spread to most European countries [51]. A 2018 survey in 14 European countries demonstrated that QX and 793B were the most prevalent strains [52]. The Middle East registered 43.66% of 793B cases between 2009 and 2014 [53], and Asia registered a high prevalence of the strain, especially in Iran (42.8%) [54]. The strain was named “G”, but in 2004 sequencing, it was shown that the variant was genetically linked to the 4/91 genotype [12,55]. Since then, 793B and Massachusetts (Mass.) have been the most prevalent serotypes worldwide [56]. In a broader context, IBV registered an important genetic diversity worldwide, with the most common serotypes being Mass-type, 4/91 (793B or CR88)-like, D274-like (D207, D212 or D1466, D3896), D3128, QX-like, and Italy02 [57]. While the Mass. and Italy02 genotypes were prevalent in Morocco between 2010 and 2014, these genotypes were not detected during this study, highlighting a shift in the distribution of strains in Morocco [12]. The same trend was seen in Europe, where the Italy02 genotype, prevalent in 2023, has declined since then [58]. In China, while the QX genotype has increased, vaccine-like genotypes such as Mass. decreased from 50.4% to 4.4% [59]. The shifts are mainly caused by a change in poultry management, viral mutations and recombination, and vaccination practices [60,61]. Phylogenetic clustering revealed a clear distinction among Moroccan isolate genotypes, with no evidence of other lineages circulating. The Moroccan IBV strains clustered closely with European and Middle Eastern isolates, especially with the 793B and D181 lineages. These detections suggest a recent introduction via the poultry trade with European countries [62], or a late detection due to the late initiation of sequencing efforts [63]. Remarkably, the D181 strain was detected for the first time in Morocco and Africa. First reported by Molenaar et al. in 2020, D181 is considered a new serotype and the second lineage within genotype II (GII) [64,65]. Phylogenetic analysis showed 90.9–95% similarity with D181 strains isolated in Dutch layers (strain CK/NL/D181/2018) [65].

In summary, various respiratory pathogens are circulating in Moroccan broiler farms, with the prevalence of IBV and H9N2 playing a key role in exacerbating multiple infections and compromising the immune system of chickens. Despite extensive monitoring and vaccination efforts, the presence of several viruses has reduced vaccine efficacy and complicated effective strategy implementation for controlling viral spread and mitigating disease severity. These findings suggest that the current vaccination strategies do not provide sufficient protection. The vaccination limitations may be attributed to various factors: antigenic variation between vaccine strains, improper vaccine administrations, or co-infections with other pathogens that could enhance viral pathogenicity. These factors compromise chickens’ immune response and optimize viral spread [66]. Therefore, vaccination alone does not provide sufficient protection. The need to regularly monitor field strains, adapt vaccination programs, and instill good health management strategies could help reduce poultry losses [66]. To achieve better goals, it is essential to implement massive standardized vaccination strategies, control flock density, and improve biosecurity practices [67].

Interactions between different viral respiratory pathogens play a crucial role in determining the outcome of coinfections. However, most studies on viral interference in complex respiratory infections fail to fully capture the field conditions, where chickens are usually exposed to multiple infectious agents simultaneously [32,38,39,68].

This study underscores the severity of viral coinfections that resulted in high morbidity and mortality rates. Respiratory diseases are complex and deserve more attention as they threaten the poultry industry. The high prevalence of viral avian diseases in Morocco can be attributed to poor sanitary conditions and inadequate vaccination programs. Our findings emphasize the urgent need to implement effective sanitary measures to limit the emergence of new variants and the further spread of viruses across the country. This approach should focus on updating vaccination programs to cover circulating genotypes, strengthen biosecurity measures to reduce viral introduction, improve vaccine administration, and reduce stress factors to decrease chickens’ susceptibility to infections [66].

## Figures and Tables

**Figure 1 viruses-17-00786-f001:**
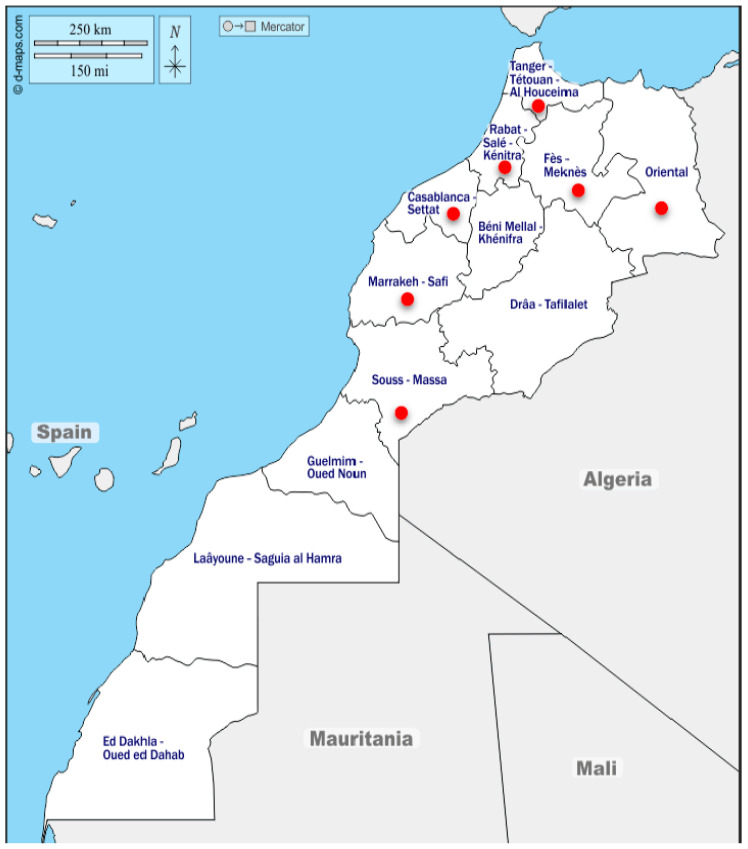
Regions of study on respiratory pathogens (H9N2, IBV, ILT, MG, and MS) in broiler chicken between 2021 and 2023. The red dots represent the regions where field investigations were conducted.

**Figure 2 viruses-17-00786-f002:**
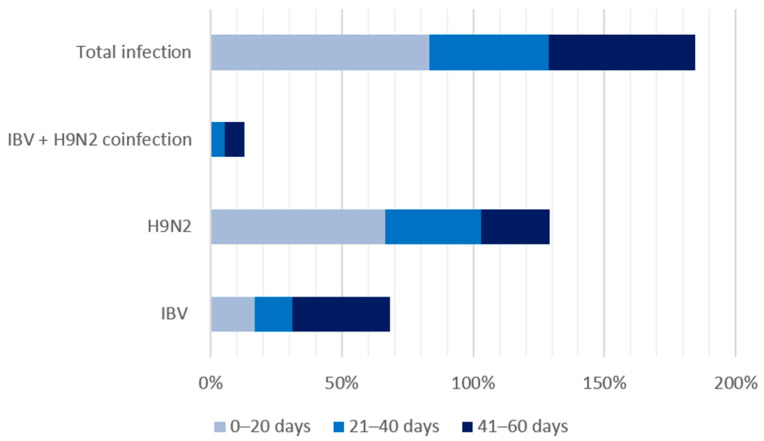
Detection rates of IBV, H9N2, and coinfections by age group.

**Figure 3 viruses-17-00786-f003:**
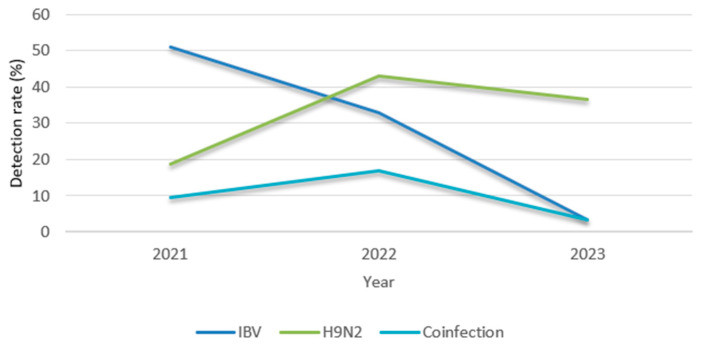
Yearly distribution of IBV, H9N2, and coinfection (2021–2023).

**Figure 4 viruses-17-00786-f004:**
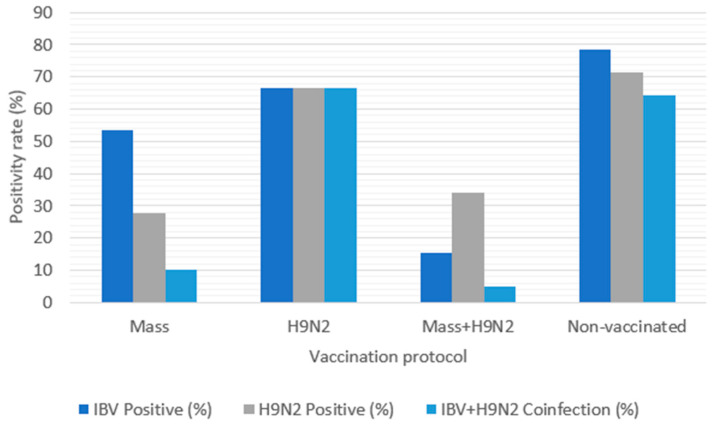
Infection and coinfection rates by vaccination protocol.

**Figure 5 viruses-17-00786-f005:**
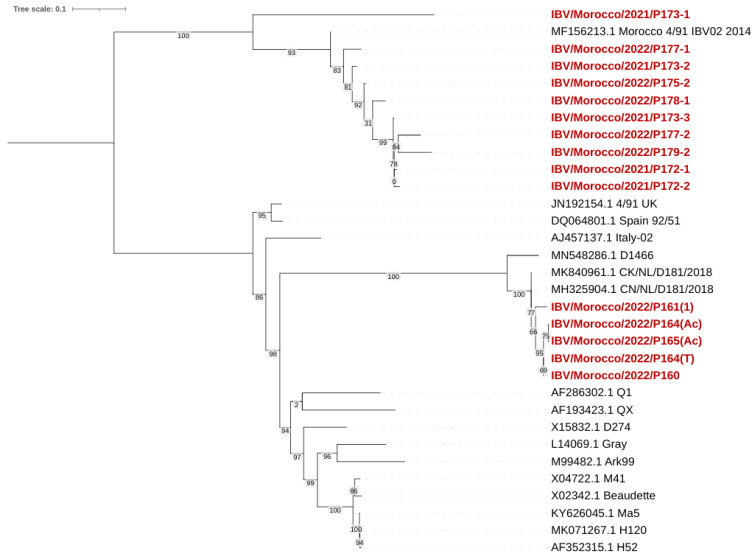
Phylogenetic tree of Moroccan IBV strains based on S1 gene sequences. Moroccan viruses sequenced for the purpose of this study are marked in red.

**Figure 6 viruses-17-00786-f006:**
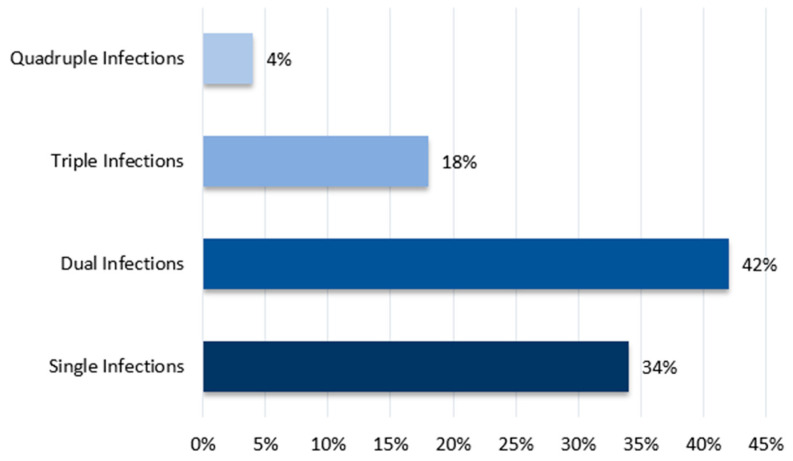
Detection rates for single, dual, triple, and quadruple infections across farms.

**Figure 7 viruses-17-00786-f007:**
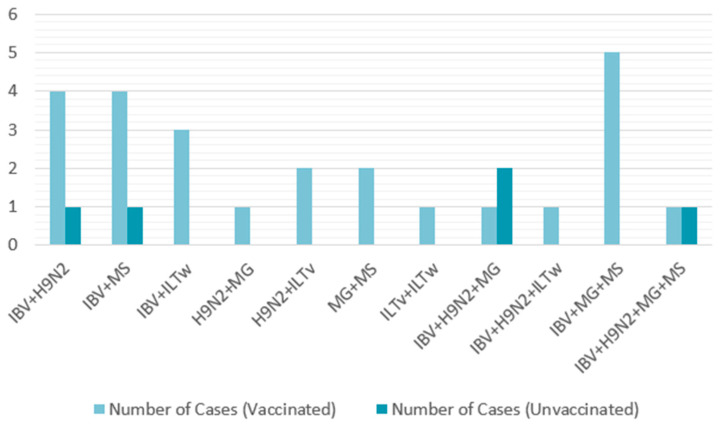
Coinfection combinations in vaccinated and unvaccinated broiler farms (2021–2023).

**Table 1 viruses-17-00786-t001:** Real-time RT-PCR primer and probe sequences for viral pathogens.

Target	Primer/Probe	Sequence (5′ to 3′)	References
IBV	IBV-Forward	GCTTTTGAGCCTAGCGTT	[17,18]
	IBV-Reverse	GCCATGTTGTCACTGTCTATTG	
	IBV probe	CACCACCAGAACCTGTCACCTC	
H9N2	H9-Forward	ATGGGGTTTGCTGCC	[19,20]
	H9-Reverse	TTATATACAAATGTTGCAC(T)CTG	
	H9 probe	TTCTGGGCCATGTCCAATGG	
	N2-Forward	GCATGGTCCAGYTCAAGYTG	
	N2-Reverse	CCYTTCCAGTTGTCTCTGCA	
ILTv	ILT-Forward	CAGCTCGAAGTCTGAAGAGACA	[21]
	ILT-Reverse	AGCGAGCATACTAGGGAAACGGT	
	ILT Probe	FAM-TGAGCGGCTTCATAACATAGGATCGA	
ILTw	ILT-Forward	CAGCTCGAAGTCTGAAGAGACA	
	ILT-Reverse	GCTGCACGCCAACTCCTATG	
	ILT Probe	Cy5-TGTGCGGGTGAAACCGTA AATTACG	

FAM = fluorescein amidite; Cy5 = cyanine.

**Table 2 viruses-17-00786-t002:** Distribution of IBV and H9N2 in broiler chickens by age group (*n* = 88).

Virus	0–20 Days	21–40 Days	41–60 Days
IBV	1/6 (16.67%)	8/55 (14.55%)	10/27 (37.04%)
H9N2	4/6 (66.67%)	20/55 (36.36%)	7/27 (25.93%)
IBV + H9N2 coinfection	0/6 (0%)	3/55 (5.45%)	2/27 (7.41%)
Total infection	5/6 (83.33%)	25/55 (45.45%)	15/27 (55.56%)

**Table 3 viruses-17-00786-t003:** Distribution of IBV and H9N2 from 2021 to 2023.

Virus	2021	2022	2023
IBV	27/53 (50.94%)	33/100 (33%)	1/30 (3.33%)
H9N2	10/53 (18.87%)	43/100 (44%)	11/30 (36.67%)
IBV + H9N2 coinfection	5/53 (9.43%)	17/100 (17%)	1/30 (3.33%)
Total infection	32/53 (60.38%)	59/100 (59%)	11/30 (36.67%)

**Table 4 viruses-17-00786-t004:** Positivity for IBV, H9N2, and coinfection of IBV and H9N2.

Vaccination Protocol	Total Vaccinated	IBV Positive	H9N2 Positive	IBV + H9N2 Coinfection
IBV Mass-type vaccine	58/183 (31.7%)	31/58 (53.45%)	16/58 (27.59%)	6/58 (10.34%)
H9N2	3/183 (1.64%)	2/3 (66.67%)	2/3 (66.67%)	2/3 (66.67%)
IBV Mass-type vaccine + H9N2	103/183 (56.28%)	16/103 (15.53%)	35/103 (33.98%)	5/103 (4.85%)
Non-vaccinated	14/183 (7.65%)	11/14 (78.57%)	10/14 (71.43%)	9/14 (64.29%)

**Table 5 viruses-17-00786-t005:** Number of coinfection pathogens in vaccinated farms in Morocco from 2021 to 2023.

Number of Coinfections	Number of Infected Cases	Infection Combinations	Number of Infection Combination Cases
2	17	IBV + H9	4
		IBV + MS	4
		IBV + ILTw	3
		H9N2 + MG	1
		H9N2 + ILTv	2
		MG + MS	2
		ILTv + ILTw	1
3	7	IBV + H9N2 + MG	1
		IBV + H9N2 + ILTw	1
		IBV + MG + MS	5
4	1	IBV + H9N2 + MG + MS	1

**Table 6 viruses-17-00786-t006:** Number of coinfection pathogens in unvaccinated farms in Morocco from 2021 to 2023.

Number of Coinfections	Number of Infected Cases	Infection Combinations	Number of Infection Combination Cases
2	2	IBV + H9N2	1
		IBV + MS	1
3	2	IBV + H9N2 + MG	2
4	1	IBV + H9N2 + MG + MS	1

**Table 7 viruses-17-00786-t007:** Regional distribution of pathogens in farms across Morocco from 2021 to 2023.

Number of Pathogens	Region	Coinfections	Number of Farms	Total Farms
4	Rabat-Sale-Kenitra	IBV + H9N2 + MG + MS	1	2
	Casablanca-Settat	IBV + H9N2 + MG + MS	1	
3	Rabat-Sale-Kenitra	IBV + MG + MS	1	9
		IBV + H9N2 + MG	2	
	Casablanca-Settat	IBV + MG + MS	4	
		IBV + H9N2 + MG	1	
	Marrakech-Safi	IBV + H9N2 + ILTw	1	
2	Rabat-Sale-Kenitra	IBV + H9N2	5	19
		IBV + MS	2	
		IBV + ILTw	3	
		MG + MS	1	
	Casablanca-Settat	IBV + MS	1	
		H9N2 + MG	1	
		H9N2 + ILTw	1	
		MG + MS	2	
		ILTv + ILTw	1	
	Marrakech-Safi	H9N2 + ILTw	1	
	East	IBV + MS	1	

## Data Availability

Data are contained within the article.

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
