# Peer review of "Surveillance and Coinfection Dynamics of Infectious Bronchitis Virus and Avian Influenza H9N2 in Moroccan Broiler Farms (2021–2023): Phylogenetic Insights and Impact on Poultry Health"

_viruses, 2025, doi:10.3390/v17060786_

Round 1

Reviewer 1 Report

Comments and Suggestions for Authors

The authors have provided a comprehensive description of the surveillance and coinfection dynamics of infectious bronchitis virus (IBV) and avian influenza H9N2 in Moroccan broiler farms. Monitoring co-infections of pathogens in poultry flocks is crucial for systematically analyzing disease dynamics. The manuscript is well-written and clearly demonstrates the virus circulation in the broiler farms. However, some missing information needs to be added to make the paper more complete and robust.

  1. The title mentions phylogenetic insights of the pathogens, but no phylogenetic tree of the pathogens was constructed in the manuscript. The phylogenetic trees of the IBV and H9N2 viruses need to be added to the paper.
  2. The specific vaccine information (e.g., strain, manufacturer, and batch number) for the chicken flocks vaccinated against the tested pathogens, especially for the IBV and H9N2, should be listed in the Materials and Methods section.
  3. Prevention strategies (e.g., vaccination schedules, biosecurity) should be discussed in the context of findings.
  4. The entire manuscript should be thoroughly proofread to eliminate any spelling or formatting errors.

Example:

The term 'RT' in Line 21 should not be formatted in bold.

The sentence in line 175 is confusing to readers. 'Farms t received both the Mass and H9N2 vaccines had a significantly lower positivity rate of 33.98%.' The use of 't' in this context is unclear to readers. Please clarify its meaning.

Comments on the Quality of English Language

The authors should carefully review the manuscript to improve the clarity and professionalism of the English language.

Author Response

  1. The title mentions phylogenetic insights of the pathogens, but no phylogenetic tree of the pathogens was constructed in the manuscript. The phylogenetic trees of the IBV and H9N2 viruses need to be added to the paper.

Thank you for your valuable suggestion. We fully agree with the reviewer’s observation. We would like to clarify that the primary objective of this study is to investigate the circulation and epidemiological dynamics of Infectious Bronchitis Virus (IBV) in Moroccan broiler farms. Accordingly, the phylogenetic analysis was specifically focused on IBV strains, and we have included a phylogenetic tree to highlight the genotypic diversity, with particular emphasis on the emergence of the D181 genotype in Morocco.

While H9N2 was also detected in the studied flocks—as a frequent co-infecting virus that may act synergistically with IBV—it was identified using real-time PCR only, such as others respiratory pathogens. Sequencing and phylogenetic analysis were conducted exclusively for IBV.

Nevertheless, in response to the reviewer’s suggestion, we will include appropriate references and a brief overview of the current understanding of H9N2 genotypic diversity in Morocco within the revised manuscript.

  1. The specific vaccine information (e.g., strain, manufacturer, and batch number) for the chicken flocks vaccinated against the tested pathogens, especially for the IBV and H9N2, should be listed in the Materials and Methods section.

We appreciate the reviewer’s comment and understand the importance of including specific vaccine information. However, we regret to inform you that we do not have access to detailed vaccine data (e.g., strain, manufacturer, and batch number) for the flocks included in this study. This is because the investigation focused on suspected field cases of IBV, and such information was not disclosed by the supervising veterinarians or farm managers. Despite our efforts, these details remain unavailable.

  1. Prevention strategies (e.g., vaccination schedules, biosecurity) should be discussed in the context of findings.

Thank you for your comment. The reviewer is correct, and the relevant information has now been incorporated into the discussion section (lines 406-417) as follows: Despite extensive monitoring and vaccination efforts, the presence of several viruses has reduced vaccine efficacy and complicated effective strategies for controlling viral spread and mitigating disease severity. The vaccination limitations may be due to various factors: antigenic variation between vaccine strains, improper vaccine administrations, or co-infections with other pathogens that could enhance viral pathogenicity. These factors compromise chickens’ immune response and optimize viral spread [65]. Therefore, vaccination alone doesn’t provide sufficient protection. The need to regularly monitor field strains, adapt vaccination programs, and instill good health management strategies could help reduce poultry losses [65]. To achieve better goals, it is essential to implement massive standardized vaccination strategies, control flock density, and improve biosecurity practices [66].

  1. The entire manuscript should be thoroughly proofread to eliminate any spelling or formatting errors.

Thank you for pointing this out. We have carefully proofread the entire manuscript to eliminate any spelling, grammatical, and formatting errors.

  1. Comments on the Quality of English Language

The authors should carefully review the manuscript to improve the clarity and professionalism of the English language.

Thank you for your valuable feedback. We have thoroughly revised the manuscript to improve the clarity, grammar, and overall quality of the English language. We have ensured that the text meets a high standard of professionalism and readability throughout.

Reviewer 2 Report

Comments and Suggestions for Authors

Dear Authors

Thank you for the opportunity to review the interesting work by Regragui et al, manuscript title “Surveillance and coinfection dynamics of infectious bronchitis virus and avian influenza H9N2 in Moroccan broiler farms (2021-2023): phylogenetic insights and impact on poultry health”. The current study on the surveillance of multiple pathogens in Moroccan boiler farms provides epidemiologically important information regarding the current microbial threat from IBV, avian influenza, and other pathogens.  The data looks interesting, but incomplete. There is enormous scope to improve the overall quality of the manuscript.  Please find my comments below.

  • Significant improvements are needed in the representation of the graphs or tables. Authors should include a better representation with a pie diagram and a stacked bar chart. To represent the data.
  • “Phylogenetic analysis of S gene identified two main IBV genotypes: 793B and D181, with the latter being a strain is circulating for the first time in Moroccan poultry”. There is no phylogenetic tree in the manuscript. It is unclear how many IBV samples were sequenced to generate the phylogenetic tree. The author must provide information and include phylogenetic trees as the main figure in the manuscript.
  • Authors also stated that the main objective of the study is to identify the current IBV genotypes and lineages circulating in the country. They must include other currently circulating strains (from different geographic locations) of IBV to construct a phylogenetic tree. They should try to explain the emergence of new genotypes based on a phylogenetic tree, not just sequence alignments. The manuscript requires a comprehensive picture of circulating IBV strains worldwide and in Morocco. 
  • In Table 1, only H9 primers were included. What about N2 primers? Authors must clarify how they confirmed the H9N2 subtypes of influenza.
  • “Non-vaccinated farms recorded the highest positivity rate (78.57%), while those vaccinated with the Mass vaccine recorded a positivity rate of 53.45%”. What is the mass vaccine? Clarify the breadth and targeted pathogens intended to be protected through vaccination.
  • Authors must present figures based on pathogen-specific vaccine status, timeline, and presence or absence of specific infection.
  • Figure 2. Positivity rates for respiratory pathogens.” It is very difficult to understand the goal of this figure. What are positive and negative samples? Need clarification, the figure legend should indicate that.
  • “Co-infected flocks exhibited severe clinical signs and lesions, especially between 21-40 days of age.” No data is provided to back this statement. Authors may think of including the clinical and lesion data.
  • “This study underscores the severity of viral coinfections, which resulted in high morbidity and mortality rates”. The manuscript doesn't convey the outcome of the co-infection. How do those infections lead to culling or mortality? Authors need to provide quantitative data.
  • Authors must discuss whether vaccinations are protective, and if not, possible reasons contributing to high morbidity and mortality.

I am looking forward to the revised manuscript.

Thank you

Author Response

  1. Significant improvements are needed in the representation of the graphs or tables. Authors should include a better representation with a pie diagram and a stacked bar chart. To represent the data.

We appreciate the reviewer’s suggestion. We will revise our figures and include pie diagrams and stacked bar charts to present the data more clearly and comprehensively.

  1. “Phylogenetic analysis of S gene identified two main IBV genotypes: 793B and D181, with the latter being a strain is circulating for the first time in Moroccan poultry”. There is no phylogenetic tree in the manuscript. It is unclear how many IBV samples were sequenced to generate the phylogenetic tree. The author must provide information and include phylogenetic trees as the main figure in the manuscript.

Thank you for this important observation. We acknowledge the omission of the phylogenetic tree in the submitted version of the manuscript. In response to the reviewer’s comment, we have now included the phylogenetic tree of the S1 gene as a main figure in the revised manuscript. This tree illustrates the classification of the identified IBV strains into two main genotypes: 793B and D181, with the latter reported here for the first time in Moroccan poultry.

Additionally, we have added detailed information in the Materials and Methods section specifying the number of IBV-positive samples that were sequenced and used in the phylogenetic analysis, along with a clear description of the sequencing and tree construction methods.

  1. Authors also stated that the main objective of the study is to identify the current IBV genotypes and lineages circulating in the country. They must include other currently circulating strains (from different geographic locations) of IBV to construct a phylogenetic tree. They should try to explain the emergence of new genotypes based on a phylogenetic tree, not just sequence alignments. The manuscript requires a comprehensive picture of circulating IBV strains worldwide and in Morocco.*

Thank you for your insightful comment. In the revised manuscript, we have updated the phylogenetic analysis to include reference IBV strains representing major genotypes currently circulating in different geographic regions worldwide, as well as previously reported Moroccan strains. This broader context allows for a more comprehensive comparison and strengthens our ability to characterize the genetic relationships and divergence patterns observed in our isolates.

We have also revised the Results and Discussion sections to provide a more detailed explanation of the emergence of the newly identified D181 genotype in Morocco. This analysis is now based on phylogenetic clustering, rather than solely on sequence alignments, and includes a comparative discussion of our findings with those reported in international studies (lines 231-243) in Results, and (lines 373-401) in Discussion. This comprehensive approach enhances our understanding of the evolutionary relationships between the Moroccan strains and globally circulating IBV genotypes, thereby reinforcing the study’s objective of identifying and contextualizing the current IBV genotypes circulating in the country.

  1. In Table 1, only H9 primers were included. What about N2 primers? Authors must clarify how they confirmed the H9N2 subtypes of influenza.

We thank the reviewer for raising this important point. In our study, confirmation of the H9N2 subtype was performed using a two-step molecular approach. First, initial screening was conducted using H9-specific primers targeting a conserved region of the HA gene, as  listed in Table 1. Second, N2 subtyping was performed by RT-PCR using a primer pair specific for the N2 gene segment, as described by Fereidouni et al. (2008). We will include the N2 primer sequences and relevant reference in the revised Table 1 and Materials and Methods section for clarity.

  1. “Non-vaccinated farms recorded the highest positivity rate (78.57%), while those vaccinated with the Mass vaccine recorded a positivity rate of 53.45%”. What is the mass vaccine? Clarify the breadth and targeted pathogens intended to be protected through vaccination.

Thank you for your comment. The term "Mass vaccine" refers specifically to the Massachusetts (Mass) strain-based vaccine used for protection against Infectious Bronchitis Virus (IBV). To avoid any confusion, we will revise the manuscript to clearly state "IBV Mass-type vaccine" and clarify that this vaccine is intended solely for the prevention of IBV infections.

  1. Authors must present figures based on pathogen-specific vaccine status, timeline, and presence or absence of specific infection.

We appreciate the reviewer’s suggestion and agree with the importance of presenting clearer data. In response, we have included new figures 2, 3, 4, 6, and 7 that specifically illustrate the pathogen-specific vaccine status, vaccination timelines, and the presence or absence of infections in the study population.

  1. Figure 2. Positivity rates for respiratory pathogens.” It is very difficult to understand the goal of this figure. What are positive and negative samples? Need clarification, the figure legend should indicate that.

Thank you for this comment. The figure has been deleted and other more appropriate figures have been added to the manuscript.

  1. “Co-infected flocks exhibited severe clinical signs and lesions, especially between 21-40 days of age.” No data is provided to back this statement. Authors may think of including the clinical and lesion data.

Thank you for this valuable comment. We acknowledge that the statement regarding clinical signs and lesions in co-infected flocks was not fully supported by data in the original manuscript. In response, we have now included detailed data on the clinical signs and lesions observed in co-infected flocks across all age groups (lines 24-25). 17 infections were recorded in vaccinated chickens aged 20-40 days, with infection peaks at 24 and 38 days, indicating a higher susceptibility in this age range (lines 282-283).

  1. “This study underscores the severity of viral coinfections, which resulted in high morbidity and mortality rates”. The manuscript doesn't convey the outcome of the co-infection. How do those infections lead to culling or mortality? Authors need to provide quantitative data.

Thank you for pointing this out. We recognize that the manuscript did not fully convey the impact of co-infections on poultry health. In response, we have now included quantitative data on morbidity and mortality rates observed in co-infected flocks:

 -paragraph 3.4.1: Weekly mortality rates ranged from 0.1% to 16.4%.

 -paragraph 3.4., The most prevalent clinical signs observed included abnormal mortality in 44.26% of farms

While direct data on culling were not recorded, these mortality figures likely reflect both disease-associated deaths and farmer-driven culling in response to severe clinical signs.”

Discussion on Vaccination Efficacy

Authors must discuss whether vaccinations are protective, and if not, possible reasons contributing to high morbidity and mortality.

Thank you for your valuable suggestion. In response, we have expanded the Discussion section to include a detailed analysis of the protective efficacy of vaccination (lines 404-416). We now address potential factors that could contribute to reduced vaccine effectiveness, such as antigenic variation, co-infections, and issues related to improper vaccine administration. These factors may explain the observed high morbidity and mortality rates despite vaccination efforts.

Round 2

Reviewer 2 Report

Comments and Suggestions for Authors

Dear Authors

Thank you for addressing my concerns and suggestions. I am satisfied with the revised manuscript. 

Thank you

Comments on the Quality of English Language

Dear Authors 

I am satisfied with the language.

Thank you